# A Single Center Observational Study on Clinical Manifestations and Associated Factors of Pediatric Long COVID

**DOI:** 10.3390/ijerph20186799

**Published:** 2023-09-21

**Authors:** Enrica Mancino, Raffaella Nenna, Luigi Matera, Domenico Paolo La Regina, Laura Petrarca, Elio Iovine, Greta Di Mattia, Antonella Frassanito, Maria Giulia Conti, Enea Bonci, Mattia Spatuzzo, Sara Ialongo, Anna Maria Zicari, Alberto Spalice, Fabio Midulla

**Affiliations:** 1Department of Translational and Precision Medicine, Sapienza University of Rome, 00161 Rome, Italy; 2Department of Maternal, Infantile and Urological Sciences, Sapienza University of Rome, 00161 Rome, Italy; 3Department of Experimental Medicine, Sapienza University of Rome, 00161 Rome, Italy

**Keywords:** COVID-19, Long COVID, children, persistent symptoms

## Abstract

Children with SARS-CoV-2 are mostly mild symptomatic, but they may develop conditions, such as persisting symptoms, that may put them at greater risk of complications. Our aim was to evaluate the frequency and the presence of risk factors for persisting COVID-19 symptoms in children. We carried out a prospective observational study of the clinical manifestation of Long COVID at the Department of Maternal Infantile Science of a tertiary University hospital in Rome. We included 697 children (0–18 years), with previous SARS-CoV-2 infection. Children and parents were asked questions regarding persistent symptoms of COVID-19. Children with symptoms 30 days after initial diagnosis were 185/697 (26.4%). Moreover, 81/697 (11.6%) patients presented symptoms 90 days after the diagnosis. Thirty-day-persisting symptoms were mostly present in children with anosmia, atopy, asthenia, and cough in the acute phase compared with the asymptomatic children 30 days after infection. After 90 days, symptoms described were mainly neurological (47/697 children, 6.7%), and headache (19/697; 2.7%) was the most frequent manifestation. In conclusion, a relatively large proportion of the patients reported persisting symptoms that seem to be related to the symptom burden and to the atopy. Ninety days after the infection, most of the children had recovered, showing that long-term effects are not frequent. Limitations of the study include the single-center design and the lack of a control group.

## 1. Introduction

The World Health Organization (WHO) declared coronavirus disease 19 (COVID-19) a global pandemic in March 2020 [1], and the direct or indirect impact on the life of every child is still very deep. SARS-CoV-2 is known to cause a wide spectrum of diseases, from common colds and self-limiting upper respiratory tract infections to severe systemic diseases [2].

To date (May 2023), according to the COVID-19 integrated surveillance data in Italy, there have been 4,844,180 confirmed cases of COVID-19 among patients from 0 to 19 years of age, including 26,139 hospitalized, 587 admitted to the pediatric intensive care unit (PICU) and 90 deaths [3]. 

Worldwide, the research community published a large amount of data regarding COVID-19 and soon found out that children mostly experience mild respiratory symptoms with sporadic complications [4]. 

The most common symptoms in children are fever, cough, and sore throat [5]; other frequent symptoms include diarrhea, vomiting, headache, fatigue, myalgia, and nasal congestion [6]. Although children are frequently paucisymptomatic or asymptomatic during the acute phase of the disease, they may develop conditions after the initial SARS-CoV-2 infection that may put them at greater risk of complications. Multi-inflammatory syndrome of children (MISC) and Long COVID are the long-term consequences of SARS-CoV-2 infection of more interest. MISC is an immune-mediated disease that occurs in a small proportion (<0.1%) of children two to six weeks after being infected with SARS-CoV-2 [7,8,9]. 

Long COVID refers to a range of new, returning, or ongoing symptoms after initial SARS-CoV-2 infection. Different definitions of Long COVID have already been proposed, and recently the WHO defined it as a condition that occurs in individuals with a history of SARS-CoV-2 infection, usually three months from the onset of COVID-19, with symptoms that last for at least two months and cannot be explained by an alternative diagnosis [10]. 

While the definition is not univocal, the scientific community agrees that Long COVID is present also in children. Buonsenso et al. [11] followed 129 children who were diagnosed with microbiologically confirmed COVID-19. 52.7% of patients had at least one symptom that persisted for four months (120 days) or more. The most frequently reported symptoms were: insomnia (18.6%), respiratory and chest symptoms (14.7%), nasal congestion (12.4%), fatigue (10.8%), muscle pain (10.1%), joint pain (6.9%), and concentration difficulties (10.1%). The authors described that although the Long COVID symptoms were more frequent in the patients who were originally symptomatic in the acute phase, there was a significant proportion of patients who reported persisting symptoms despite being asymptomatic during the acute SARS-CoV-2 infection. Parents’ perspectives about the persistent symptoms were investigated in a survey developed by *LongCOVIDKids* that showed the persistence of symptoms for an average of 8.2 months [12]. J Ludvigsson investigated Long COVID in a case report of five children and adolescents who had persistent symptoms for a period of six to eight months. At follow-up, the children showed a similar set of symptoms: fatigue, dyspnea, heart palpitations, or chest pain. Four out of the five children also described headaches, difficulties in concentration, muscle weakness, dizziness, and a sore throat. Other reported symptoms included abdominal pain, memory loss, depression, myalgia, and skin rashes. Two children experienced: remitting fever, sleep disorders, joint pain, diarrhea, vomiting, and hyper anesthesia. One child suffered from a poor appetite, a chronic cough, numbness, and a persisting alteration of the sense of smell and taste [13]. The effects on the quality of life (QoL) are very important but a case-control study suggests that persistent symptoms could be associated with factors other than SARS-CoV-2 infection, including psychosocial factors [14]. 

Children can experience Long COVID, but, to date, few studies include a large group of children and very little is known about possible clinical sequelae. In this scenario, it becomes crucial to identify long-term potential consequences in children and the relationship with the acute illness. 

The purpose of this study was to document the prevalence, symptoms, and risk factors of ‘long COVID’ in a large cohort of SARS-CoV-2 infected children <18 years and to give a better understanding of the potential long-term consequences in children.

## 2. Materials and Methods

### 2.1. Study Population

We conducted an observational prospective, single-center, study at the Department of Maternal Infantile Science of a tertiary University hospital in Rome. The Department has introduced an outpatient service for pediatric patients with a SARS-CoV-2 infection at least 30 days before the visit, and, within this programme, we consecutively enrolled 697 patients aged 0 to 18 years from February 2021 to November 2021 (Figure 1). 

Patients aged 0 to 18 years with a SARS-CoV-2 infection at least 30 days before the visit were included. Patients >18 years old or children with neurocognitive disability were excluded to avoid an improper assessment of signs and symptoms included in the questionnaire.

### 2.2. Data Collection

The infection was documented by a positive nasopharyngeal swab (reverse transcription PCR or rapid antigen test) for emerging symptoms or previous contact with positive subjects. Both symptomatic and asymptomatic patients during the acute infection were enrolled. 

During the visit, demographic and clinical data such as age, gender, breastfeeding, cigarette smoke exposure, body weight, and gestational age were systematically collected. Children were classified as underweight, normal weight, overweight, and obese according to the Body Mass Index (BMI). Based on the Expert Committee Recommendations, underweight is defined as a BMI below the 5th percentile; overweight is defined as a BMI in the 85th to less than the 95th percentile; obesity is defined as a BMI at or above the 95th percentile for children and teens of the same age and sex [15]. During the visit, caregivers were interviewed about their child’s health using a structured questionnaire (Appendix A) for evaluation of acute and persisting symptoms. Persistent symptoms were defined as those extending for at least 30 days after diagnosis, which cannot be explained by an alternative diagnosis.

Participants were categorized into groups, according to the presence of symptoms during the acute disease (subsequently mentioned as “symptomatic children during SARS-CoV-2 infection”) and according to the presence or the absence of symptoms 30 days after the diagnosis (subsequently mentioned as “symptomatic children 30 days after infection” and “asymptomatic children 30 days after infection). As headaches are a frequently reported COVID-19 symptom, patients with these symptoms will be also analyzed separately. Caregivers of patients with symptoms 30 days after COVID-19 diagnosis were subsequently interviewed using the same structured questionnaire by three pediatricians by phone. Parents were asked about the presence of new, returning, or ongoing COVID-19 related symptoms 90 days after the onset of COVID-19. Informed consent was obtained from all subjects involved in the study. The local Ethics Committee approved the study protocol, and informed parental consent was obtained from all patients (RIF.CE 0399/2021).

### 2.3. Statistical Analysis

According to the Kolmogorov–Smirnov test, none of the continuous variables were normally distributed. Their values were expressed as median and range. Categorical variables were expressed as numbers and percentages and were compared using appropriate nonparametric tests (chi-square test, Kendall tau-B, etc.). 

The binary logistic regression model was conducted to identify associations of the independent variables with symptomatic and asymptomatic children 30 days (and 90 days—data not shown) after SARS-CoV-2 infection (dichotomous) with respect to the study situation during the acute phase of the disease. We used forward and stepwise procedures to confirm that the results were stable and generalizable, independent of the model approach used. The independent variables included: sex, preterm, atopy, previous bronchiolitis, previous wheezing, obesity, positive swab test >21 days, asthenia, cough, and fever. Nonsignificant variables were excluded stepwise via forward elimination and dropped at the level of *p* < 0.05.

All the analyses were carried out using IBM SPSS software version 27 (SPSS software, IBM Corporation, NY, USA).

## 3. Results

Six hundred and ninety-seven children with previous SARS-CoV-2 infection were enrolled (364 (52.2%) males; median age 9.6 years; range: 0.1–18.6). General characteristics are shown in Table 1.

Symptoms during COVID-19 are shown in Table 2.

Regarding symptoms 30 days after SARS-CoV-2 infection, 185/697 (26.4%) children were symptomatic. 15/697 patients reported exercise-induced dyspnea (2.2%), 10/697 cough (1.4%), and 5/679 patients manifested dyspnea (0.7%) 30 days after infection.

Gastrointestinal symptoms were reported by 16/697 (2.3%) patients, and reduction of food intake, nausea, and increase of food intake were present respectively in 7/697 (1.0%), 3/697 (0.4%) and 2/697 (0.3%) children. Ninety-four out of six hundred and ninety-seven (94/697, 13.5%) children had neurological and neurobehavioral symptoms: anosmia (34/697; 4.9%), headache (32/697; 4.6%) and ageusia (17/697, 2.4%) were the most frequent neurological manifestations. Asthenia was present in 86/697 (12.3%) patients (Table 2).

Age groups reported symptoms with the following frequencies expressed as a percentage; 12.7% (0–4 years old), 19.3% (5–10 years old), 38.7% (11–15 years old), 45.0% (16–18 years old), data not shown. 

Analyzing symptomatic and asymptomatic children 30 days after SARS-CoV-2 infection, patients with symptoms more frequently had a past history positive for atopy (60/185, 31% vs. 114/512, 22%, *p*-value = 0.006). The percentage of overweight and obese females was statistically higher in symptomatic patients than in asymptomatics (34/185, 18% vs. 62/512, 12%, *p* = 0.03). The 185 symptomatic patients more frequently have had symptoms (172/185, 93% vs. 384/512, 76%, *p*-value < 0.001), fever (106/185, 58% vs. 240/512, 47%, *p*-value = 0.01), five or more symptoms (69/185, 38% vs. 64/512, 12%, *p*-value < 0.001) and asthenia (76/185, 41% vs. 104/512, 20%, *p*-value < 0.001) during SARS-CoV-2 infection than children without symptoms 30 days after infection (Table 3a).

A significant correlation between symptoms 30 days after COVID-19 diagnosis and anosmia, atopy, asthenia, and cough was confirmed by the logistic regression analysis (Table 3b).

The binary logistic regression model indicated that anosmia, atopy, asthenia, and cough were significant predictors of symptomatic and asymptomatic children 30 days after SARS-CoV-2 infection (Table 3b). The other selected predictors—sex, preterm, previous bronchiolitis, previous wheezing, obesity, positive swab test >21 days, fever—were not significant. 

The findings most strongly associated with symptomatic and asymptomatic children 30 days after the SARS-CoV-2 infection are COVID symptoms (anosmia, asthenia, and cough), but also atopy seems to play an important role (Logistic analysis is included in the added results).

By dividing our population by the percentile of BMI, a headache was reported more frequently by overweight and obese patients than by underweight and normal weight children both during the infection and 30 days after the infection (*p*-value = 0.007 during the infection, and *p*-value = 0.04 for Headache 4 weeks after infection; using Kendall’s Tau B) (Figure 2).

Symptomatic patients 90 days after infection and for at least two months were 81/697 (11.6%). Among the 13/697 (1.9%) patients with respiratory manifestations, the symptoms reported were dyspnea (11/697; 1.6%), rhinitis (5/697; 0.7%), and cough (3/697; 0.4%). Gastrointestinal symptoms were reported by 5/697 (0.7%) patients and abdominal pain, weight loss, and nausea were present respectively in 3/697 (0.4%), 2/697 (0.3%), and 2/697 (0.3%) children. 47/697 (6.7%) had neurological symptoms and headache (19/697; 2.7%), anosmia (16/607; 2.3%) and lack of concentration (14/697; 2.0%) were the most frequent neurological manifestations. Asthenia was present in 39/697 (5.6%) patients (Table 2).

Regarding patients with symptoms 90 days after infection, symptomatic patients were more frequently preterm (7/81, 8.6% vs. 9/104, 8.6%, *p*-value = 0.04), exposed to passive smoking (24/81, 30% vs. 22/104, 21%, *p*-value 0.02) and they had more frequently asthenia (45/81, 56% vs. 14/104, 13%, *p*-value = 0.02) than asymptomatic patients (Table 4).

No significant correlation was found by the logistic regression analysis (data not shown).

## 4. Discussion

This large prospective observational cohort study provides evidence of persisting symptoms after SARS-CoV-2 infection. We also sought to identify demographic and clinical features associated with the onset of persisting symptoms. 

Most patients reported symptoms during SARS-CoV-2 infection, but the clinical course was predominantly mild, and only two children were diagnosed with MISC. These data on the frequency of symptomatic/asymptomatic infection in the pediatric age group correspond to previously published research [16,17]. In contrast with the first studies reporting a significant proportion of children with severe or fatal disease [18], we concluded that severe infections are rare in the pediatric population, which is concordant with what has been reported in the latest literature [19]. The exact mechanisms are yet to be defined, but children may manifest less severe disease due to a stronger innate immune response to SARS-CoV-2, a more robust antibody response, an altered distribution and activity of ACE2 receptors, the “trained immunity” from viruses or vaccines, and also fewer comorbidities with respect to adults [20]. 

Asthenia, headache and anosmia/ageusia were the most commonly reported symptoms. Less common symptoms were chest pain, arthralgia/myalgia, abdominal pain/diarrhea and dyspnea. Atopy was more frequent among symptomatic vs. asymptomatic children 30 days after infection. The association between related COVID-19 symptoms and atopy is much debated in the literature, but the vast majority of studies indicate that allergic diseases do not represent a risk factor for COVID-19 susceptibility nor cause a more severe course of disease [21,22]. In our study, the percentage of subjects suffering from atopy is higher in symptomatic patients, but it is also close to the percentage of atopy in the general population. The average BMI centile was calculated as being higher in the symptomatic group of females compared with the asymptomatic group of children. Obesity has been previously alluded to as a risk factor for disease severity and mortality [23], and this finding highlights the importance of a healthy lifestyle and a balanced diet for children. 

As headaches have been reported very frequently, we sought the link between this important symptom of COVID-19 and psycho-social status and obesity. It appears that overweight and obese children have reported this symptom more often. So headaches are more common in overweight and obese children, both during the infection and at a distance from the infection: we could say that it is a feature that increases with weight. Previous research advises that children who suffer from headaches follow a correct lifestyle and that this is of the utmost importance in childhood, as it will improve the quality of life. Mechanisms for the obesity-headache association are not known. It is possible that the disorders share similar pathophysiological pathways and/or that lifestyle and behavioral factors contribute to this relationship [24,25,26]. We also assume that the stress and uncertainty surrounding the pandemic has been more prevalent among children with psychosocial issues, and overweight and obese children are included in this category [27].

Upon the comparison between acute phase symptoms and the frequency of symptoms 30 days after the infection, persisting symptoms were more frequent among the children with anosmia, atopy, asthenia, and cough in the acute phase. The symptom burden may correlate with persisting symptoms. This finding has been alluded to in a study by Sudre et al. following persisting symptoms in adults [28]. We hypothesize that, upon further research, the symptom burden may be predictive of long-term symptoms in children as well. Of all of the other risk factors and comorbidities, we found no significant increase in the frequency of persisting symptoms. 

After 90 days most symptoms were resolved, and symptoms described were mainly neurological. This confirms that the evolution of the disease is mainly mild in children, and long-term effects are infrequent. We also suspect that the psychological effects have a significant contribution towards the subsequent persisting symptoms. Psychosomatic effects may be triggered by the stress involved with a global pandemic. Deconditioning may also be an issue due to the increased amount of time spent indoors during the quarantine. Brusaferri et al. present preliminary evidence of pandemic-related neuroinflammation in non-infected participants, providing an example of how broad the impact of the pandemic has been on human health, extending beyond the morbidity directly induced by the virus itself [29]. Moreover, in a large nationwide study documenting long COVID in children, 0.8% of SARS-CoV-2-positive children reported symptoms lasting >4 weeks, when compared to a control group. The authors concluded that symptoms such as concentration difficulties, headache, muscle- and joint pain as well as nausea are not ‘long COVID’ symptoms but they could be symptoms reflecting psychological and social consequences of the pandemic [30]. This evidence is consistent with our idea that part of the reported symptoms could be assigned to psychological sequelae of social restrictions. However, the exact mechanism underlying the persisting symptoms is not fully understood, and the presence of inflammation may be responsible, at least in part, for the persisting symptoms [31]. 

Limitations of the study include the single-center design, but we included a large sample size. We only collected data so a possible bias in the estimation of persisting symptoms exists and without a control group, it is difficult to compare the frequency of reported symptoms with the general population. Another potential bias due to the lack of a control group should be related to the presence of nonspecific long COVID reported symptoms that are prevalent in the general population. Moreover, information on the presence of symptoms 90 days after infection was only requested from those who reported symptoms 30 days after COVID-19 diagnosis. We then lost 30-day asymptomatic patients who became symptomatic later on, but we believe that they represent a small and unremarkable group of patients.

## 5. Conclusions

In conclusion, a relatively large proportion of the children reported persisting symptoms following SARS-CoV-2 infection. A large component of these persisting symptoms are subjective symptoms. We suspect that psychological factors have a large role to play in the symptomatology following SARS-CoV-2 infection. This should not, however, divert clinician’s attention away from children who have been infected with SARS-CoV-2. This is not only because severe manifestations such as MIS-C are possible, but also because of the potential psychological consequences following the COVID-19 pandemic. Children’s mental health may still suffer as a result of infection due to fear of the unknown consequences and the stigma attached to infection. Deconditioning is also likely to contribute to persisting symptoms in children, as they have been unable to go to school and, during the full lockdown, were confined indoors.

## Figures and Tables

**Figure 1 ijerph-20-06799-f001:**
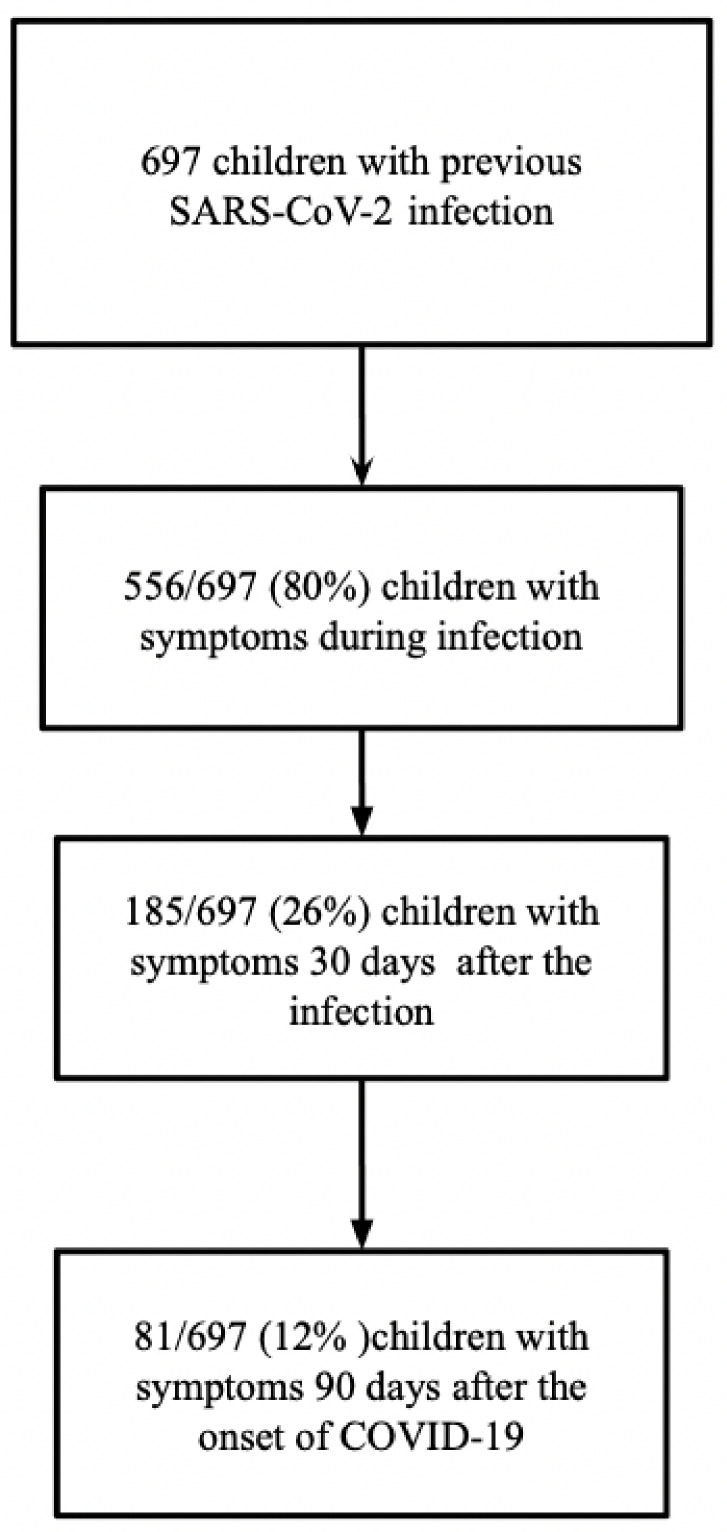
Flow chart of study population.

**Figure 2 ijerph-20-06799-f002:**
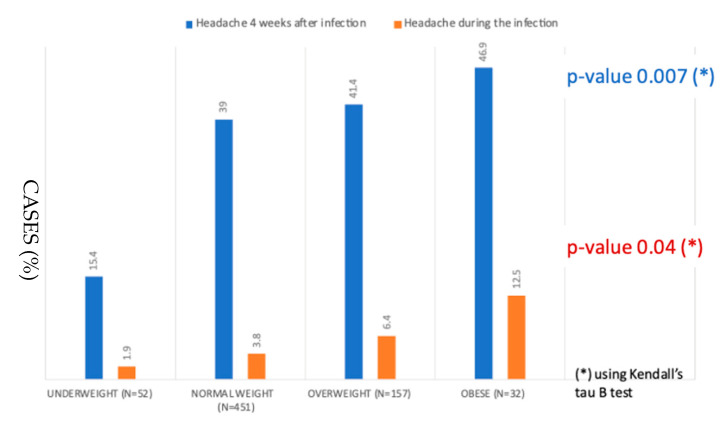
Percentage of cases with headache during the infection and 30 days after the infection in underweight, normal weight, overweight and obese children.

**Table 1 ijerph-20-06799-t001:** General characteristic of patients with previous SARS-CoV-2 infection.

	*354 Males*	*342 Females*	*697 Children*	*p-* Value
Age, years (median, range)			9.6 (0.1–18.6)	
- 0–4 years (no, %)	65 (18)	45 (13)	110 (16)	
- 5–10 years (no, %)	152 (43)	156 (46)	306 (44)	0.314
- 11–15 years (no, %)	117 (33)	121 (35)	236 (34)	
- 16–18 years (no, %)	20 (5.8)	20 (5.6)	40 (5.7)	
Preterm (<37 weeks) (no, %)	41 (55)	33 (45)	74 (11)	0.395
Gestational age, weeks (median, range)	39 (29–42)	39 (25–42)	39 (25–42)	0.539
Breastfeeding (no, %)	280 (51)	271 (49)	551 (79)	0.779
Smoking exposure (no, %) *	119 (51)	113 (49)	232 (33)	0.373
Previous bronchiolitis	73 (57)	54 (43)	127 (18)	0.103
Previous wheezing	91 (56)	71 (44)	162 (23)	0.133
Atopy ^$^	95 (55)	79 (45)	174 (25.0)	0.284
Body Mass Index (centile)				
- Underweight - Normal weight - Overweight - Obesity	31 (60) 227 (50) 72 (46) 21 (66)	21 (40%) 224 (50) 85 (54) 11 (34)	52 (7.5) 451 (65) 157 (22) 32 (4.6)	0.112
Persistently positive COVID-19 patients (positive swab test > 21 days)	151 (52)	139 (48)	290 (42)	0.618
Days elapsed between first positive SARS-CoV-2 test and outpatient consultation (median, range)	174 (36–424)	174 (25–669)	174 (25–669)	0.691

* defined as parental smoking in the child’s house; ^$^ defined as allergic diseases associated with at least one skin prick test positive for inhalants, as reported by the parents.

**Table 2 ijerph-20-06799-t002:** Acute COVID-19 symptoms and COVID-19 persistent symptoms (30 and 90 days after infection) reported by patients.

	Symptomatic Children during SARS-CoV-2 *n = 697*	Symptomatic Children 30 Days after Infection *n* = 697	Symptomatic Children 90 Days after Infection *n* = 697
Symptomatic patients	556 (80%)	185 (26%)	81/697 (11.6%)
Fever ^#^	346 (50%)	0 (0.0%)	0 (0.0%)
Respiratory Symptoms - Cough - Rhinitis - Pharyngitis - Dyspnea - Otitis - Exercise induced dyspnea - Other respiratory symptoms ^1^	**281 (40%)** 154 (22%) 120 (17%) 96 (14%) 47 (6.7%) 20 (2.9%) 0 (0.0%) 9 (1.3%)	**30 (4.3%)** 10 (1.4%) 6 (0.9%) 1 (0.1%) 5 (0.7%) 0 (0.0%) 15 (2.2%) 15 (2.2%)	**13 (1.9%)** 3 (0.4%) 5 (0.7%) 0 (0.0%) 11 (1.6%) 0 (0.0%) 0 (0.0%) 0 (0.0%)
Gastrointestinal Symptoms - Diarrhea - Vomiting - Abdominal pain - Stomach pain - Nausea - Poor Appetite/weight loss - Increase in food intake	**143 (20.5%)** 102 (15%) 47 (6.7%) 17 (2.4%) 11 (1.6%) 10 (1.7%) 1 (0.1%) 0 (0%)	**16 (2.3%)** 3 (0.4%) 0 (0.0%) 4 (0.6%) 0 (0.0%) 3 (0.4%) 7 (1.0%) 2 (0.3%)	**5 (0.7%)** 0 (0.0%) 0 (0.0%) 3 (0.4%) 0 (0.0%) 2 (0.3%) 2 (0.3%) 2 (0.3%)
Neurological Symptoms - Headache * - Anosmia - Ageusia - Altered taste - Altered Smell - Lack of Concentration - Loss of memory - Sleeping disorders - Anxiety - Syncope - Dizziness - Other neurological symptoms ^2^	**339 (48.6%)** 264 (38%) 142 (20%) 141 (20%) 1 (0.1%) 1 (0.1%) 1 (0.1%) 1 (0.1%) 0 (0%) 0 (0%) 0 (0%) 4 (0.6%) 17 (2.4%)	**94 (13.5%)** 32 (4.6%) 34 (4.9%) 17 (2.4%) 11 (1.6%) 1 (0.1%) 8 (1.1%) 1 (0.1%) 12 (1.7%) 7 (1.0%) 1 (0.1%) 1 (0.1%) 0 (0.0%)	**47 (6.7%)** 19 (2.7%) 16 (2.3%) 13 (1.9%) 1 (0.1%) 0 (0.0%) 14 (2.0%) 3 (0.4%) 9 (1.3%) 5 (0.7%) 0 (0%) 0 (0%) 9 (1.3%)
Musculoskeletal symptoms - Muscle pain	**89 (13%)** 67 (9.6%)	**17 (2.4%)** 10 (1.4%)	7 (1%) 7 (1%)
Asthenia	180 (26%)	86 (12.3%)	39 (5.6%)
Skin rash	28 (4.0%)	4 (0.6%)	3 (0.4%)
Chest pain	6 (0.9%)	3 (0.4%)	3 (0.4%)
Palpitations	1 (0.1%)	2 (0.3%)	1 (0.1%)
Other ^3^	17 (2.7%)	3 (0.4%)	0 (0.0%)
≥5 symptoms during SARS-CoV-2 infection	131 (19%)	--	--
MISC ^4^	---	2 (0.3%)	---

^1^: wheezing, sinusitis, aphonia; ^2^: cold sweat, panic attack, irritability; ^3^: conjuntivitis, hair loss, lymph node enlargement, aphthous ulcers, epistaxis; ^4^: Multisystem inflammatory syndrome in children; ^#^ defined as having a temperature of 37.5 °C or greater; * defined as pain in any region of the head that lasts at least 15 min.

**Table 3 ijerph-20-06799-t003:** (a). Demographic and clinical features of symptomatic and asymptomatic children 30 days after SARS-CoV-2 infection. (b). The binary logistic regression model: association between symptomatic and asymptomatic children 30 days after the SARS-CoV-2 infection and several independent variables.

(a)										
					Symptomatic Children 30 Days after Infection *n* = 185		Asymptomatic Children 30 Days after Infection *n* = 512			*p*-Value
Sex (male)					88 (48%)		267 (52%)			*0.33*
Preterm					21 (11%)		53 (10%)			*0.68*
Breastfeeding					149 (82%)		402 (79%)			*0.40*
Smoking exposure *					66 (46%)		166 (41%)			*0.28*
Persistently positive COVID-19 patients (positive swab test > 21 days)					84 (47%)		206 (41%)			*0.18*
Previous bronchiolitis					31 (17%)		96 (19%)			*0.56*
Previous wheezing					50 (27%)		112 (22%)			*0.15*
Atopy					60 (31%)		114 (22%)			** *0.006* **
Underweight					7 (3.8%)		45 (8.9%)			*0.03*
Overweight and obese					57(30%)		132 (26%)			*0.19*
Overweight and obese (female)					34 (18%)		62 (12%)			** *0.03* **
Overweight and obese (male)					23 (12%)		70 (14%)			0.66
Obesity					21 (11%)		35 (6.8%)			0.07
Symptoms during SARS-CoV-2 infection (No,%)					172 (93%)		384 (76%)			** *<0.001* **
Fever during SARS-CoV-2 infection (No,%)					106(58%)		240 (47%)			** *0.01* **
≥5 symptoms during SARS-CoV-2 infection					69 (38%)		64 (12%)			** *<0.001* **
Asthenia (No,%)					76 (41%)		104 (20%)			** *<0.001* **
**(b)**										
**Step 5a**		**B**	**S.E.**	**Wald**	**df**	**Score**	**OR**	**95% C.I. for OR**		** *p* **
								**Lower**	**Upper**	
	Anosmia	0.864	0.210	16.952	1		2.373	1.573	3.581	**0.001**
	Atopy	0.495	0.205	5.827	1		1.640	1.097	2.450	**0.016**
	Asthenia	0.725	0.199	13.282	1		2.065	1.398	3.049	**0.001**
	Cough	0.853	0.209	16.667	1		2.346	1.558	3.532	**0.001**
	Sex				1	1.429				0.232
	Preterm				1	0.518				0.472
	Previous bronchiolitis				1	0.248				0.618
	Previous wheezing				1	0.102				0.750
	Obese				1	0.919				0.338
	Positive swab test >21 days				1	1.280				0.258
	Fever				1	0.383				0.536

***** Defined as parental smoking in the child’s house. a. Variable(s) entered on step 4: Atopy; B = Beta coefficient; S.E. = standard error; Wald = Wald test; df = degrees of freedom; OR = odds ratio; 95% C.I. = 95% confidence interval.

**Table 4 ijerph-20-06799-t004:** Demographic and clinical features of symptomatic and asymptomatic children 90 days after SARS-CoV-2 infection.

	Symptomatic Children 90 Days after Infection *n* = 81	Asymptomatic Children 90 Days after Infection *n* = 104	*p*-Value
Sex (male)	40 (49%)	22 (21%)	*0.75*
Preterm	7 (8.6%)	9 (8.6%)	** *0.04* **
Breastfeeding	67 (83%)	30 (29%)	*0.11*
Smoking exposure *	24 (30%)	22 (21%)	** *0.02* **
Persistently positive COVID-19 patients (positive swab test > 21 days)	34 (42%)	21 (20%)	*0.43*
Previous bronchiolitis	12 (15%)	7 (6.7%)	*0.78*
Previous wheezing	22 (27%)	10 (9.6%)	*0.68*
Atopy	29 (36%)	16 (15%)	*0.80*
Underweight	1 (1.2%)	4 (3.8%)	** *0.03* **
Overweight and obese	29 (36%)	12 (11%)	0.42
Overweight and obesity (males)	13 (16.0%)	7 (6.7%)	0.93
Overweight and obesity (female)	16 (20%)	5 (4.8%)	0.27
Symptoms during SARS-CoV-2 infection (N0,%)	77 (95%)	38 (36%)	*0.33*
Fever during SARS-CoV-2 infection (No,%)	49 (60%)	24 (23%)	*0.72*
≥5 symptoms during SARS-CoV-2 infection	34 (42%)	15 (14%)	*0.50*
Asthenia (No,%)	45 (56%)	14 (13%)	** *0.02* **

***** Defined as parental smoking in the child’s house.

## Data Availability

Data available on request from the authors.

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
