# Peer review of "A Single Center Observational Study on Clinical Manifestations and Associated Factors of Pediatric Long COVID"

_ijerph, 2023, doi:10.3390/ijerph20186799_

Round 1
Reviewer 1 Report (Previous Reviewer 2)
The comments are addressed and the manuscript is improved.
Good
Author Response
no comments from the reviewer
Reviewer 2 Report (Previous Reviewer 3)
The authors conducted the study “A single center observational study on clinical manifestations and associated factors of pediatric Long COVID” to document prevalence, symptoms and risk factors of ‘long COVID’ in a large cohort of SARS-CoV-2 infected children <18 years and to give a better understanding of the potential long term consequences in children.
The study has a critical weakness in its data and conclusion due to the lack of an appropriate control group. Long COVID should be strictly discriminated from general sickness (e.g. a general cold) that show similar symptoms to long COVID. Including this weakness, the reviewer would like to suggest that the manuscript may address following points before publication.
1.
In the last paragraph of Discussion, the authors wrote limitations of this study. The reviewer wonders if the authors could state more details of the potential biases due to the lack of a control group in this study in “Limitations of the study”.
Moreover, the reviewer would like to suggest the authors to add one sentence to Abstract regarding the possible bias of this study without a control group.
2.
The authors showed the result of a logistic regression analysis in Table 3b. However, Table 3b seems to be incomplete and please finalize Table 3b like other Tables.
Furthermore, not-significant values should also be shown in Table 3b or supplementary Table (such as Sex, Preterm, Previous bronchiolitis, etc.) to be compared with significant ones.
3.
Figure 2 does not have its Y axis. Please put the Y axis and add an explanation of a unit in the Y axis.
Author Response
-
In the last paragraph of Discussion, the authors wrote limitations of this study. The reviewer wonders if the authors could state more details of the potential biases due to the lack of a control group in this study in “Limitations of the study”. Thank you for your comment. We provided more details in “Limitations of the study”.Moreover, the reviewer would like to suggest the authors to add one sentence to Abstract regarding the possible bias of this study without a control group. Following your suggestions, we added limitations of the study at the end of the Abstract.
-
The authors showed the result of a logistic regression analysis in Table 3b. However, Table 3b seems to be incomplete and please finalize Table 3b like other Tables. Furthermore, not-significant values should also be shown in Table 3b or supplementary Table (such as Sex, Preterm, Previous bronchiolitis, etc.) to be compared with significant ones. According to your comment, we made some adjustments to Table 3b.
-
Figure 2 does not have its Y axis. Please put the Y axis and add an explanation of a unit in the Y axis. Thank you for your comment. We putted the Y axis and added an explanation.
Reviewer 3 Report (Previous Reviewer 4)
The manuscript "A single center observational study on clinical manifestations and associated factors of pediatric Long COVID" provides a point of reference for clinicians evaluating children having symptoms post COVID. It is a large prospective study where children with confirmed infection were enrolled at presentation and followed for symptoms. I think it is an interesting data set which merits publishing. The authors have addressed the prior review concerns adequately.
Please double check your percentages. For example, you wrote 7/81, 8.8% but it should be 8.6%.
Please verify your P values as well.
Under Results: The sentence "The other selected predictors - Sex, Preterm, Previous bronchiolitis, Previous wheezing..." has several words capitalized that shouldn't be. Please fix.
no major revisions needed
Author Response
- Please double check your percentages. For example, you wrote 7/81, 8.8% but it should be 8.6%. Please verify your P values as well.
Thank you. We checked and modified the values as required.
- Under Results: The sentence "The other selected predictors - Sex, Preterm, Previous bronchiolitis, Previous wheezing..." has several words capitalized that shouldn't be. Please fix. According to your suggestions, we made some adjustments to the sentence.
Reviewer 4 Report (Previous Reviewer 5)
The authors have improved the manuscript by addressing many of the reviewer's criticisms, however some minor points still require revision as follows:
- The sentences "Asthenia was present in 86/697 (12.3%) patients" and "Symptomatic patients 90 days after infection and for at least 2 months were 81/697 (11.6%)" do not correspond to what is shown in Table 2 (i.e., 11.6% and 12%, respectively).
- In Figure 2, the y-axis title is still missing and the legend does not explain the type of central tendency measure shown (e.g., mean, median or mode).
- It is not clear what "step" means in Table 3b.
Minor editing of English language required.
Author Response
- The sentences "Asthenia was present in 86/697 (12.3%) patients" and "Symptomatic patients 90 days after infection and for at least 2 months were 81/697 (11.6%)" do not correspond to what is shown in Table 2 (i.e., 11.6% and 12%, respectively). Thank you. We made the adjustments required.
- In Figure 2, the y-axis title is still missing and the legend does not explain the type of central tendency measure shown (e.g., mean, median or mode). Thank you for your suggestions. According to your comment we made some adjustments to Figure 2.
- It is not clear what "step" means in Table 3b. As stated in the method section, we performed the stepwise regression that is a step-by-step construction model. It involves a step-by-step selection of independent variables to be involved in the final model. Atopy is the last variable to make a significant contribution to the regression (step 4). In Table 3b we showed the final Step 5, this is why you find “step” in this table. We hope we’ve made it clear.
Reviewer 5 Report (New Reviewer)
I read it with great interest. I would like to know if the children were able to attend school or if they had to take a leave of absence.
I also thought it was a bit short-sighted to attribute this to psychological stress. I would like to see more pathological exploration, since the same kind of long covid develops in adults as well.One suggestion would be to include a comparison with adult symptoms.
I think it is very important for parents and educators to know that these symptoms also occur in children.
Author Response
- I read it with great interest. I would like to know if the children were able to attend school or if they had to take a leave of absence. Most of children with long Covid symptoms were able to attend school but some of them (mostly children with asthenia, chest pain and muscle pain) were not able to play sport for a while.
- I also thought it was a bit short-sighted to attribute this to psychological stress. I would like to see more pathological exploration, since the same kind of long covid develops in adults as well. One suggestion would be to include a comparison with adult symptoms. I think it is very important for parents and educators to know that these symptoms also occur in children. As far as we know, the mechanisms underlying the long COVID symptoms are not fully understood. We explored the pathological mechanisms of persisting symptoms in a case-control study including adolescences that contracted SARS-CoV-2 infection and healthy controls. We found that TGF-β together with BDNF were higher in post-infected-COVID-19 symptomatic and long COVID girls (Petrella C et al. Serum NGF and BDNF in Long-COVID-19 Adolescents: A Pilot Study. Diagnostics (Basel). 2022 May 7;12(5):1162. doi: 10.3390/diagnostics12051162. PMID: 35626317; PMCID: PMC9140550.). At the same time, a retrospective brain imaging study shows preliminary findings on neuroimmune activation in healthy individuals during the pandemic. Authors provided an example of how broad the impact of the pandemic has been on human health, extending beyond the morbidity directly induced by the virus itself (Brusaferri L, et al. The pandemic brain: Neuroinflammation in non-infected individuals during the COVID-19 pandemic. Brain Behav Immun. 2022 May;102:89-97). These two studies suggest that we don’t know for sure what causes long COVID. We modified the discussion part to underline this aspect.
This manuscript is a resubmission of an earlier submission. The following is a list of the peer review reports and author responses from that submission.
Round 1
Reviewer 2 Report
I read the manuscript with interest. I have the following minor comments:
1. The abstracts does not include where is the study based on (location)
2. Line 37: To date......: i\please, indicate the exact date ....
3. Line 50 and 64: Either use SARS-COV2 infection or COVID-19, but not COVID-19 infection.
4. The introduction should be separated into different paragraphs; currently the introduction presented as one paragraph. Also, consider making it shorter.
5. Include the percentages in Figure 1.
6. A ssparate heading about the variables or questionnaire is recommended in the methods rather than only study population and statistics,
Relatively appropriate
Reviewer 3 Report
The authors conducted a prospective observational study to find key factors in long COVID in children aged 0-18 years. The current study successfully identified several characteristics in children, such as atopy and BMI, related to symptoms more than 30 days after infection as risk factors for long COVID.
The manuscript is well-written and the data are impressive enough to attract the reader’s interest. The reviewer would like to suggest several minor concerns that might be addressed by the authors before publication.
1.
In Table 3b, there is no explanation for characters in Table, such as “B”, “S.E.”, “Wald”, etc. Footnotes to explain these characters would help readers understand the data, since the readers are not always familiar with logistic regression analysis.
2.
In line 174, there is a description that “(p-value 0.007) (Figure 2)” in the explanation for Fig.2. However, statistical significance is not shown in Fig.2. Please show significance appropriately in Fig.2 (between which bars did the authors obtain p=0.007?)
3.
In Table 4, p-values are shown for analysis between symptomatic and asymptomatic children 90 days after infection. The reviewer is wondering if the statistical analyses were appropriately implemented for Table 4. For example, “Symptoms during SARS-CoV-2 infection” shows the result of 95% vs 36%, but it has the p-value of 0.33; however, “Smoking exposure” shows the result of 30% vs 21% with the significant p-value of 0.02. Please check the p-values calculated in Table 4.
4.
There have been several reports on long COVID in children/adolescents in other countries. Please discuss briefly the differences of results between the current study and previous studies conducted in other countries. That may address country/region-specific characteristics of long COVID in children.
5.
There are several mistakes in spelling or font. Please check the entire manuscript again to correct these mistakes. (e.g. Table 3a, N0 should be No; Table 3a, all p-values should be italic; Table 3a&4, the font is different only in “5 symptoms during SARS-CoV-2 infection”; lines 159-161, please use dots for p-values such as 0.001; etc.)
Reviewer 4 Report
This is a single center study looking at manifestations and risk factor for pediatric Long COVID. This is an important topic that continues to affect many children and family and so more information is important to add to the current lacking body of knowledge. While single center, nearly 700 patients were enrolled during the study period, 81 of which had symptoms 90days after onset of COVI-19. The figures and tables are informative and clear to understand. The symptoms evaluated are clearly defined and described. A link between several symptoms and underlying risk factors are pointed out. The only question I have is if the degree of preterm be further characterized since gestational age can be a marker for lung disease.
Reviewer 5 Report
The manuscript reports a prospective observational, single-center, study on clinical manifestations and risk factors of long COVID in a large cohort of children with previous SARS-CoV-2 infection. Although the topic is relevant, some minor points requires revision and are listed below in the order of appearance in the text:
Lines 37-39: Indicate the geographical range of the COVID-19-related epidemiological data (i.e., national or worldwide).
Lines 43/56/66/76: Start sentences in new paragraphs to make the text easier to read.
Lines 85-89: Provide the exclusion criteria used for sampling.
Line 92: Clarify which diagnostic test the nasopharyngeal swab was subjected to (e.g., RT-qPCR).
Line 144: Correct "13%" to "13.5%" in the percentage of symptomatic children 30 days after infection with neurological symptoms, "12%" to "12.3%" in the percentage of symptomatic children 30 days after infection with asthenia and "12%" to "11.6%" in the percentage of symptomatic children 90 days after infection (Table 2).
Line 145: The superscript 2 ("stomach pain") in not indicated in Table 2.
Lines 159-161: Replace comma with period as decimal separator for p-values.
Line 164: In Table 3a, provide the percentage values exactly as indicated in the main text and format the font of "≥ 5 symptoms during SARS-CoV-2 infection".
Line 170: Explain the abbreviations used in Table 3b.
Line 175: In Figure 2, the y-axis title is missing and the legend does not explain the type of central tendency measure shown (e.g., mean, median or mode) or indicate the significant differences observed.
Line 191: In Table 4, provide the percentage values exactly as indicated in the main text and format the font of "≥ 5 symptoms during SARS-CoV-2 infection".
Lines 309-316: Conflicts of interest were not properly declared by the authors.
Minor editing of English language required.